# Advances in the Management of Medullary Thyroid Carcinoma: Focus on Peptide Receptor Radionuclide Therapy

**DOI:** 10.3390/jcm9113507

**Published:** 2020-10-29

**Authors:** Erika Grossrubatscher, Giuseppe Fanciulli, Luca Pes, Franz Sesti, Carlotta Dolci, Federica de Cicco, Annamaria Colao, Antongiulio Faggiano

**Affiliations:** 1Endocrine Unit, ASST Grande Ospedale Metropolitano Niguarda, 20162 Milan, Italy; 2NET Unit, Department of Medical, Surgical and Experimental Sciences, University of Sassari-Endocrine Unit, AOU Sassari, 07100 Sassari, Italy; 3Endocrine Clinic, Policlinico Sassarese, 07100 Sassari, Italy; lpes@policlinicosassari.it; 4Department of Experimental Medicine, Sapienza University of Rome, 00161 Rome, Italy; franz.sesti@uniroma1.it (F.S.); antongiulio.faggiano@uniroma1.it (A.F.); 5Nuclear Medicine Unit, ASST Grande Ospedale Metropolitano Niguarda, 20162 Milan, Italy; carlotta.dolci@ospedaleniguarda.it; 6Department of Clinical Medicine and Surgery, Endocrinology Unit, University Federico II, 80131 Naples, Italy; fed.decicco@studenti.unina.it (F.d.C.); colao@unina.it (A.C.)

**Keywords:** peptide receptor radionuclide therapy, medullary thyroid carcinoma, somatostatin analogues, neuroendocrine neoplasm

## Abstract

Effective treatment options in advanced/progressive/metastatic medullary thyroid carcinoma (MTC) are currently limited. As in other neuroendocrine neoplasms (NENs), peptide receptor radionuclide therapy (PRRT) has been used as a therapeutic option in MTC. To date, however, there are no published reviews dealing with PRRT approaches. We performed an in-depth narrative review on the studies published in this field and collected information on registered clinical trials related to this topic. We identified 19 published studies, collectively involving more than 200 patients with MTC, and four registered clinical trials. Most cases of MTC were treated with PRRT with somatostatin analogues (SSAs) radiolabelled with 90 yttrium (90Y) and 177 lutetium (177Lu). These radiopharmaceuticals show efficacy in the treatment of patients with MTC, with a favourable radiological response (stable disease, partial response or complete response) in more than 60% of cases, coupled with low toxicity. As MTC specifically also expresses cholecystokinin receptors (CCK2Rs), PRRT with this target has also been tried, and some randomised trials are ongoing. Overall, PRRT seems to have an effective role and might be considered in the therapeutic strategy of advanced/progressive/metastatic MTC.

## 1. Introduction

Medullary thyroid carcinoma (MTC) is a rare neuroendocrine neoplasm (NEN). Surgery, with complete resection of the tumour, is, in most cases, curative [1]. However, in a subgroup of patients, the tumour shows an aggressive behaviour. In this setting (advanced/progressive/metastatic MTC), the management remains challenging [1,2]. Among systemic therapies, tyrosine kinase inhibitors (TKIs) have recently been shown to improve progression-free survival (PFS) in patients with advanced disease [3,4]. Two TKIs, namely vandetanib and cabozantinib, have been approved for the treatment of progressive or symptomatic MTC. Both drugs are effective, resulting in partial response (PR) or stable disease (SD), but tumour escape frequently occurs. On the other hand, both drugs may cause grade III or IV adverse events (classified according to the Common Terminology Criteria for Adverse Events of the National Cancer Institute) [3,4]. Therefore, other systemic options are needed. As in other advanced/progressive/metastatic NENs, radionuclide therapy may represent a therapeutic option, and indeed, it has been applied in MTC since the 1980s. Historically, the first radiopharmaceutical used in therapy was 131Iodine (131I)-metaiodobenzylguanidine (MIBG) [5,6], but its use has been limited in MTC, due to the low percentage of MTC that shows uptake on MIBG scintigraphy [7,8,9]. 

Somatostatin receptor (SSR) expressions in MTC cells have been reported in vitro and in vivo [10,11,12,13,14,15,16]. This finding provides the basis for peptide receptor radionuclide therapy (PRRT) with radiolabelled somatostatin analogues (SSAs), and indeed, several studies on the therapeutic use of these radiopharmaceuticals in MTC have been performed in the last two decades. Additionally, the cholecystokinin 2 receptor (CCK2R) expressions in MTC [17,18,19,20] offer another promising target for PRRT.

To date, there are no published reviews dealing with the different PRRT approaches in MTC comparing the outcomes in terms of efficacy and safety.

## 2. Aim of the Study

We aimed to perform an in-depth narrative review on the studies published on PRRT and MTC, in order to offer to the physician involved in the management of MTC an updated overview of the PRRT treatments.

Furthermore, we collected information on registered clinical trials (RCTs) related to this topic, in order to give the scientific community a glance about the future of PRRT in MTC.

## 3. Materials and Methods

### 3.1. Published Studies

A search on PubMed, Embase and Cochrane was made by using the terms “medullary thyroid carcinoma”, medullary thyroid cancer”, “radionuclide”, lutetium”, “yttrium”, “indium”, “peptide receptor radionuclide therapy”, “radiolabelled somatostatin analogues”, “gastrin-like peptides” and “cholecystokinin-like peptides”. Additional studies were identified by reviewing the references of all selected articles. Only studies in the English language were included. We excluded from the analysis: (1) single-case reports, (2) abstracts from congresses and proceedings from workshops and (3) letters to the editor. The search was last updated 16 October 2020.

### 3.2. RCTs

By using the same terms used in the section describing published studies, we performed a search on the Registries Clinical Trials Gov and EudraCT. The search was last updated 16 October 2020.

## 4. Results

### 4.1. Published Studies

We found 18 PRRT studies targeting SSRs involving patients with MTC: 14 with SSA labelled with the radionuclide(s) 90 yttrium (90Y) and/or 177 lutetium (177Lu) and four with SSA labelled with 111 indium (111In). Additionally, we found one study targeting cholecystokinin receptors (CCK2Rs), employing a gastrin analogue labelled with 90Y.

PRRT with 90Y and 177Lu: Fourteen studies (publication years: 1999–2020) were found [21,22,23,24,25,26,27,28,29,30,31,32,33,34]. The cumulative number of patients with MTC treated by PRRT with radiolabelled SSA was 186. The results are summarised in Table 1. 

Disease data. The disease status at baseline was represented by progressive or inoperable or advanced diseases in 142/149 patients (95.3%) in 10 studies [22,27,30,32,34].

Treatment schedule. SSA labelled with 177Lu was employed in 72 patients (38.7%) [29,30,32,33,34]. SSA labelled with 90Y was employed in 78 patients (41.9%) [21,22,23,24,25,26,27]. SSA labelled with 90Y or 177Lu were employed in 36 patients (19.4%) [28,31]. In one study, 177Lu-1,4,7,10-tetraazacyclododecane-1,4,7,10-tetraacetic acid-octreotate (DOTATATE) was used with concomitant low-dose oral capecitabine therapy as a potential radiosensitiser [34].

Radiographic response. The response was evaluated with heterogeneous criteria. Evaluation data were available for 117 patients (62.9%) in 11 studies: 44 patients showed progressive disease (PD) (37.6%), 64 patients showed SD (54.7%), six patients showed PR (5.1%), and three patients showed complete response (CR) (2.6%) [21,22,23,24,25,26,29,30,32,33,34]. 

Biochemical response. The calcitonin measurement was available for 101 patients (54.3%) in five studies [25,26,27,33,34]. In the four studies employing calcitonin plasma concentrations, evaluation data were available in 70 patients (37.6%): after PRRT, six patients showed values < 15 pg/mL (8.6%), 20 patients showed a decrease ≥ 50% in basal values (28.6%), and the remaining 44 patients showed lower reduction or increase in basal values (62.8%) [25,26,33,34]. Another study reported a post-PRRT prolongation of calcitonin doubling time ≥ 100% in 18/31 patients (58.1%) [27].

Symptomatic response and Quality of life (QoL). Data regarding the subjective response were nearly absent. The evaluation of the symptomatic response was available for 53 patients (28.5%) in two studies: the first one reported “no improvement” in all 10 patients (100%) [32]; the second one reported PD (new symptoms or symptoms increase > 30%) in 21 patients (49%), SD (symptoms reduction < 30%) in two patients (4%), PR (symptoms reduction 30–75%) in 12 patients (28%), and CR (symptoms reduction 100%) in eight patients (19%) [33]. Only one study specifically evaluated the QoL through the Short Form-36 Health Survey (SF-36) questionnaire on a small case series of six patients; all the responders (patients who had at least some tumour shrinkage) had an improvement in QoL six to 12 months after therapy [29].

Outcome. Data about time to progression (TTP) and/or PFS and/or overall survival (OS) were reported in five studies [22,26,31,32,33]. A longer survival was reported in patients with disease control (PR + SD) in the study by Salavati et al. [31] and with biochemical response in the study by Iten et al. [27]. Finally, in the study by Parghane et al. [33], a longer OS was observed in patients with longer calcitonin doubling time. 

Toxicity. Adverse reactions included two cases of treatment discontinuation (“kidney toxicity”) among 154 patients (1.3%) [21,22,23,24,25,26,27,28,29,32,33,34], one case of grade III-IV nephrotoxicity among 133 patients (0.8%) [21,22,23,24,26,27,28,29,32,33,34], and nine cases of grade III-IV haemotoxicity among 139 patients (6.5%) [21,24,25,26,27,28,29,32,33,34]. Moreover, one case of haemoptysis was reported, presumably due to the progression of pulmonary metastases [32]. No other grade III-IV treatment-related adverse events were reported in 143 patients [21,22,24,25,26,27,29,32,33,34].

PRRT with 111In: Four studies (publication years: 2000–2004) were found [35,36,37,38]. The cumulative number of patients with MTC treated by 111In-SSA was 10. The results are summarised in Table 2.

Disease data. In all cases, the disease status at baseline was represented by progressive or inoperable disease [35,36,37,38].

Radiographic response. Data about the radiologic response, evaluated with heterogeneous criteria, were available for eight patients (80.0%) in three studies: overall, four patients showed PD (40.0%), two patients showed SD (25.0%), and two patients showed CR (25.0%) [36,37,38].

Biochemical response. This parameter was evaluated with heterogeneous criteria. The evaluation of calcitonin was available for seven patients (70.0%) in three studies: after PRRT, one patient showed “normalisation” (14.2%), three patients showed “SD” (42.9%), and three patients showed “PD” (42.9%) [35,36,38].

Symptomatic response and QoL. The evaluation of the symptomatic response was available for five patients (50.0%) in the study of Valkema et al. [36]: only one patient (20.0%) showed symptomatic improvement, but the evaluation criteria were not specified. Only the study of Pasieka et al. specifically evaluated the QoL [38]: the single patient studied showed no improvement.

Outcome. Data about PFS and OS were not available in the studies.

Toxicity. Information about the adverse reactions included no cases of treatment discontinuation among four patients [35,37,38], no cases of grade III-IV nephrotoxicity among 10 patients [35,36,37,38], no cases of grade III-IV haemotoxicity among four patients [35,37,38], and no other grade III-IV treatment-related adverse events [35,36,37,38].

PRRT targeting CCK2Rs: Only one study, including eight patients, was found (publication year: 2002) [17]. Results are summarised in Table 3. 

Disease data. In all cases, the disease status at baseline was represented by progressive and advanced disease. 

Radiographic response. Since neither imaging procedures nor response criteria were specified, these data were not included in Table 3. However, patient responses were defined by the authors as follows: two PD (25.0%), four SD (50.0%), and two PR (25.0%).

Biochemical response, symptomatic response, QoL, and outcome. No data were available. 

Toxicity. Information about adverse reactions included no cases of treatment discontinuation, no cases of grade III-IV nephrotoxicity, and three cases of grade III-IV haemotoxicity (37.5%).

### 4.2. RCTs 

We identified four RCTs on ClinicalTrials.gov, two on SSRs, and two on CCK2Rs. Details of the RCTs are reported in Table 4.

## 5. Discussion

The largest part of cases of MTC reported in our review were treated with PRRT with SSAs radiolabelled with 90Y and 177Lu. The radiographic response showed a significant disease control rate (DCR = CR + PR + SD) in 62.4% of cases (CR in 2.6%, PR in 5.1%, and SD in 54.7%), most of which were in progression at the baseline. PD was observed in 37.6% of cases [21,22,23,24,25,26,29,30,32,33,34]. In two [22,30,32] of the three studies [22,30,32] in which the percentage of PD was higher, the population under study was of older age. This is in-line with the evidence that age is an important prognostic determinant in MTC [39].

The use of a concomitant radiosensitising agent, like low-dose oral capecitabine, could be even more beneficial, as suggested by a study from Satapathy et al. [40] in which the DCR was 86%. These results are in-line with similar observations made in gastroenteropancreatic NENs and malignant phaeocromocytomas about the combination of PRRT and chemotherapy [41,42,43]. 

In responsive patients, PRRT seems to stabilise the disease. However, the effects of the treatment on PFS and OS rates have yet to be determined. Indeed, only five studies evaluated these outcome endpoints. In particular, one study demonstrated longer survival in patients with a radiological disease control [31], and two studies [27,33] demonstrated longer survival in patients with a biochemical response, evaluated through calcitonin assessment, which, in MTC, can indicate tumour recurrence or progression [44] and indicate a treatment response with good sensitivity [45,46]. 

The TKI vandetanib and cabozantinib have been shown to improve PFS in patients with advanced MTC [3,4] and have been approved for the treatment of progressive or symptomatic MTC. However, both are associated in many patients with grade III-IV adverse events [3,4], which reduce the QoL of patients. Data from our review showed that a treatment discontinuation for toxicity was observed in only 1.3% of cases, grade III-IV haemotoxicity in 6.5% of cases, grade III-IV nephrotoxicity in 0.8%, and other grade III-IV side effects in no cases. The apparently higher rates of haemotoxicity compared with nephrotoxicity were derived from the high percentage of haemotoxicity found in a study from Bodei et al. [47], in which a transient lymphocytopenia was demonstrated in 88% of cases. These data could be explained by the fact that the series included patients who had received bone radiotherapy with the consequence of a potential reduction of bone marrow reserve [47]. On the contrary, the low nephrotoxicity that emerged from the data of the present review may be linked to the duration of follow-up. Indeed, it is unknown whether late renal damage may become evident four or five years after treatment administration [47].

PRRT has been used mainly in the treatment of gastroenteropancreatic NENs expressing SSRs, with promising results. Following the publication of the Neuroendocrine Tumors Therapy (NETTER-1) study [48], PRRT with 177Lu-oxodotreotide was approved by the Food and Drug Administration, European Medicines Agency, and National Institute for Health and Care Excellence in SSR-positive, well-differentiated (G1-G2) gastroenteropancreatic NENs. As for MTC, the guidelines of the European Society for Medical Oncology [49] and American Thyroid Association [50] agree that current experiences about PRRT with radiolabelled SSA in MTC are limited, and no randomised clinical trials have been carried out to compare the efficacies of PRRT versus TKIs. Consequently, at this time, PRRT should only be considered in selected patients in the context of clinical trials. However, its potential benefit when TKIs are contraindicated is recognised. 

In order to establish the role of PRRT in the therapeutic algorithm of advanced/progressive/metastatic MTC, more solid data from randomised controlled studies with survival endpoints are needed. Nevertheless, the rarity and the long natural history of the disease have hindered the research in this field.

As with other NENs, 111In-diethylenetriaminepentacetic acid (DTPA)-octreotide was the first therapeutic radiopharmaceutical used as a PRRT in MTC. The contrasting results and the limited number of patients with MTC treated do not allow a judgment on this radiopharmaceutical, which was soon abandoned, due to the low rate of objective tumour responses and the significant haematologic toxicity in favour of SSAs labelled with the more efficient 90Y and 177Lu radionuclides. Moreover, the only RCT we found, namely NCT00002947, was terminated early (reason not stated).

The limitations of this review study about data on PRRT targeting SSRs in MTC are as follows: first, most analysed case series are small, and prospective studies are few. Second, most of the data—including treatment schedule, follow-up, response evaluation, and treatments before PRRT—are highly heterogeneous and sometimes incomplete. Finally, the outcome data are available only in a few studies, due to the small sample sizes and limited lengths of follow-up, which underpowered the evaluations of PFS and OS.

New data with the primary outcome of the tumour response are expected in 2022 (RCT NCT04106843).

As previously mentioned, a PRRT could also target CCK2Rs, which are overexpressed in more than 90% of MTCs [17,18,19,20]. In the past two decades, a variety of CCK2-related peptides radiolabelled with 90Y/177Lu have been synthesised and categorised based on the sequence of their parent peptide (CCK or gastrin). To date, most studies about PRRT targeting CCK2Rs were preclinical trials and investigated specific basic criteria of candidate radioligands such as nanomolar receptor affinity, rapid and efficient accumulation in the tumour, low uptake and rapid wash-out in/from normal tissues, and in vivo stability. Only a few pilot clinical studies have been carried out in small cohorts of patients. An important landmark study in this area was the study conducted in 2002 by Behr and Behe [17], which investigated 45 patients with metastatic MTC with 111In-DTPA-minigastrin scintigraphy (23 with known and 32 with occult diseases). The radiotracer physiological uptake was related to CCK2R-specific binding and to the route of excretion. As for the binding, the normal uptake mainly regarded the stomach and, to a lower extent, gallbladder and breasts. Regarding the route of excretion, the kidneys were identified as the only excretory organs. All tumour manifestations known by conventional imaging were visualised, and at least one lesion was detected in 29/32 patients with occult disease. Furthermore, eight patients with advanced metastatic disease were injected in a dose-escalation study with potentially therapeutic activities of a 90Y-labelled minigastrin derivative at four–six-weekly intervals (Table 3, Results: section PRRT targeting CCK2Rs); haematologic and renal were identified as dose-limiting toxicities. 

In general, although very promising in a theoretical context, the clinical translation of PRRT targeting CCK2Rs has been prevented by two major problems: high kidney uptake and/or fast in vivo degradation of the different radiopeptides developed. Radiopharmaceutical research has tried to overcome these limitations by introducing various modifications in the peptide sequence of gastrin or CCK-related peptides, with the aim of reducing kidney retention and improving metabolic stability. A co-injection of plasma expanders has been proposed to reduce kidney uptake and retention, consequently minimising radiation-induced nephrotoxicity [19,20,51]. In order to reduce in vivo radiopeptide degradation, a co-injection of neutral endopeptidase inhibitors is also under investigation, since endopeptidases are the major enzymes implicated in the catabolism of gastrin/CCK-based peptides, and they are widely distributed in the body [52,53]. Research in this field is actively ongoing, as evidenced by the two clinical trials pertaining to PRRT in CCK2Rs mentioned above (Table 4); this lesser-known but promising therapy is expected to establish a new and efficient strategy for the treatment of MTC, which is still a challenging entity.

## 6. Conclusions

On the basis of the reported efficacy and low toxicity, PRRT could represent a therapeutic option in the treatment of advanced/progressive/metastatic MTC. As in other NENs, most of the experience was obtained with PRRT targeting SSRs, but CCK2Rs might offer in the future an alternative PRRT option.

## Figures and Tables

**Table 1 jcm-09-03507-t001:** PRRT targeting SSRs with 90Y and 177Lu-SSA in MTC.

References				Demographics			Disease Data			Treatment Schedule and Follow-Up				
Authors	Year	Type of Study	Subjects *n*	Age-Years Mean (Median) (Range)		F/M *n*	Mutation Status *n*	Disease Status at Baseline	Subjects with Metastases %	Radioisotope	Dose/Cycle GBq	Cycles *n*	Cumulative Dose-GBq Mean (Median) (Range)	Follow-Up Months Mean (Median) (Range)
Otte et al. [21]	1999	NS	2	64.5 (64.5) (62–67)		2/0	NS	NS	2 (100%)	90Y	1.6–2.9	4	9.4 (9.4) (9.2–9.6)	2 (2) (2–2)
Waldherr et al. [22]	2001	NS	12	55.8 (60.0) (24–72)		5/7	1 MEN2 11 NS	Progressive	NS	90Y	NS	1–4	8.1 (9.1) (1.7–14.0)	NS
Paganelli et al. [23]	2001	NS	3	51.0 (55.0) (34–64)		2/1	NS	NS	2 (67%)	90Y	1.8	3	5.5 (5.5) (5.5–5.5)	11 (12) (10–12)
Bodei et al. [24]	2003	NS	8	45.4 (46.5) (31–67)		1/7	NS	Progressive in 3/8 patients	8 (100%)	90Y	2.9–4.8	2	7.9 (8.5) (5.9–9.6)	19 (21) (4–26)
Bodei et al. [25]	2004	Retrospective	21	51.4 (53.0) (31–78)		8/13	NS	Progressive	21 (100%)	90Y	2.2–5.1 (max)	2–8	10.7 (10.4) (7.6–19.2)	(3–40)
Gao et al. [26]	2004	NS	1	57.0		0/1	NS	Progressive	1 (100%)	90Y	3.3	4	13.2	NS
Iten et al. [27]	2007	Prospective	31	(56.7) (24–77)		10/21	2 MEN2 (1 additional MEN1), 28 NS	Progressive	31 (100%)	90Y	3.7 (GBq/m^2^)	1–5	(12.6) (1.7–29.6)	(12) (1–107)
Budiawan et al. [28]	2013	NS	8	59.1 (61.0) (40–76)		4/4	NS	Progressive	8 (100%)	90Y and/or 177Lu	NS	1–3	NS	NS
Vaisman et al. [29]	2015	Prospective	7	NS		NS	NS	Progressive	NS	177Lu	7.4	4	29.6 (29.6) (29.6–29.6)	(8–12)
Lapa et al. [30]	2015	Retrospective	4	49.0 (44.0) (33–75)		2/2	2 MEN2 2NS	NS	4 (100%)	177Lu	NS	2–5	31.4 (31.4) (23.7–39.0) 2 patients NS	NS
Salavati et al. [31]	2016	NS	28	47.9 (26–72)		14/14	NS	NS	28 (100%)	90Y and/or 177Lu	NS	≤ 5	NS	NS
Beukhof et al. [32]	2019	Retrospective	10	(63.0) (19–75)		6/4	6 sporadic 4 NS	Progressive in 8/10 patients	10 (100%)	177Lu	NS	4 (mean)	(27.8–29.6)	NS
Parghane et al. [33]	2020	Retrospective	43	(48.0) (25–80)		8/35	NS	Progressive	43 (100%)	177Lu	5.5 (mean)	1–6	18.5 (5.55–33.3)	(20) (8–78)
Satapathy et al. [34]	2020	Retrospective	8	46.0 (47.5) (22–70)		5/3	8 sporadic	Progressive or advanced or inoperable	8 (100%)	177Lu *	6.0–7.4	1–4	19.1 (20.9) (6.4–27.8)	40 (34) (14–69) 1 patient NS
-	1999–2020	-	186	-		67/112	-	Progressive or Advanced or Inoperable 142/149 (95.3%)	166/167 (99.4%)	-	-	-	-	-
**References**	**Radiographic Response**						**Biochemical Response**			**Treatment Outcome**		**Treatment Toxicity**		
**Authors, Year**	**Subjects Suitable for Evaluative *n***	**Response Criteria**	**PD *n* (%)**	**SD *n* (%)**	**PR *n* (%)**	**CR *n* (%)**	**Subjects** **Suitable for Evaluative *n***	**Response Criteria (Calcitonin, CT)**	**Response**	**PFS Months**	**OS Months**	**Discontinuation *n* (%)**	**Grade III/IV Nephrotoxicity *n* (%)**	**Grade III/IV Haemotoxicity *n* (%)**
Otte et al., 1999 [21]	2	NS	0 (0%)	2 (100%)	0 (0%)	0 (0%)	0	-	-	NS	NS	0 (0%)	0 (0%)	0 (0%)
Waldherr et al., 2001 [22]	12	WHO	7 (58%)	5 (42%)	0 (0%)	0 (0%)	0	-	-	NS (TTP: mean 8, median 10, range 3–14)	NS	0 (0%)	0 (0%)	NS
Paganelli et al., 2001 [23]	3	WHO	0 (0%)	3 (100%)	0 (0%)	0 (0%)	0	-	-	NS	NS	0 (0%)	0 (0%)	NS
Bodei et al., 2003 [24]	7	WHO	2 (29%)	3 (43%)	1 (14%)	1 (14%)	0	-	-	NS	NS	0 (0%)	0 (0%)	7 (88%)
Bodei et al., 2004 [25]	21	SWOG	7 (33%)	12 (57%)	0 (0%)	2 (10%)	21	PD (increase ≥ 25% in basal value) SD (none of the others) PR (decrease ≥ 50% in basal value) CR (<15 pg/mL)	12 (57%) 3 (14%) 5 (24%) 1 (5%)	NS	NS	0 (0%)	NS	1 (5%)
Gao et al., 2004 [26]	1	WHO	0 (0%)	1 (100%)	0 (0%)	0 (0%)	1	Pre-therapy and Post-therapy values	10,461 pg/mL; 3414 pg/mL	NS (TTP: 6)	NS	0 (0%)	0 (0%)	0 (0%)
Iten et al., 2007 [27]	0	-	-	-	-	-	31	Post-PRRT Prolongation of CT Doubling Time (≥100%)	Response = 18/31 (58%)	NS	16 (median) 1-107 (range)	2 (6%): kidney toxicity	1 (3%)	1 (3%)
Budiawan et al., 2013 [28]	0	-	-	-	-	-	0	-	-	NS	NS	0 (0%)	0 (0%)	0 (0%)
Vaisman et al., 2015 [29]	7	RECIST 1.1	1 (14%)	3 (43%)	3 (43%)	0 (0%)	0	-	-	NS	NS	0 (0%)	0 (0%)	0 (0%)
Lapa et al., 2015 [30]	4	RECIST 1.1 “in most cases”	4 (100%)	0 (0%)	0 (0%)	0 (0%)	0	-	-	NS	NS	NS	NS	NS
Salavati et al., 2016 [31]	0	-	-	-	-	-	0	-	-	NS	24 in PD (median) 36 in SD (median) 72 in PR (median)	NS	NS	NS
Beukhof et al., 2019 [32]	10	RECIST 1.1	6 (60%)	4 (40%)	0 (0%)	0 (0%)	0	-	-	8 (median) 4–144 (range)	14 (median) 5–144 (range)	0 (0%)	0 (0%)	0 (0%)
Parghane et al., 2020 [33]	43	RECIST 1.1	16 (37%)	25 (58%)	2 (5%)	0 (0%)	43	PD (increase ≥ 30% in basal value) SD (none of the others) PR (decrease ≥ 50% in basal value) CR (<15 pg/mL)	21 (49%) 4 (9%) 13 (30%) 5 (12%)	24 (median)	26 (median)	0 (0%)	0 (0%)	0 (0%)
Satapathy et al., 2020 [34]	7	RECIST 1.1	1 (14%)	6 (86%)	0 (0%)	0 (0%)	5	Increase ≥ 25% in basal value None of the others Decrease ≥ 50% in basal value CT < 15 pg/mL	2 (40%) 2 (40%) (20%) 0 (0%)	NS	NS	0 (0%)	0 (0%)	0 (0%)
-	117	-	44 (37.6%)	64 (54.7%)	6 (5.1%)	3 (2.6%)	101	-	-	-	-	2/154 (1.3%)	1/133 (0.8%)	9/139 (6.5%)

PRRT: peptide receptor radionuclide therapy; SSR: somatostatin receptor; 90Y: 90 yttrium; 177Lu: lutetium; SSA: somatostatin analogue; MTC: medullary thyroid carcinoma; F: female; M: male; GBq: GigaBecquerel; NS: not specified; MEN: multiple endocrine neoplasia; PD: progressive disease; SD: stable disease; PR: partial response; CR: complete response; PFS: progression-free survival; OS: overall survival; WHO: World Health Organization; TTP: time-to-progression; SWOG: Southwest Oncology Group; RECIST: Response Evaluation Criteria In Solid Tumours. * Concomitant low-dose oral capecitabine therapy (1.250 mg/m^2^/day from day 0 to day 14 of each PRRT therapy).

**Table 2 jcm-09-03507-t002:** PRRT targeting SSRs with 111In-SSA in MTC.

References				Demographics			Disease Data			Treatment Schedule and Follow-Up				
Authors	Year	Type of Study	Subjects *n*	Age-Years Mean (Median) (Range)		F/M *n*	Mutation Status *n*	Disease Status at Baseline	Subjects with Metastases %	Radioisotope	Dose/Cycle GBq	Cycles *n*	Cumulative Dose-GBq Mean (Median) (Range)	Follow-Up Months Mean (Median) (Range)
Caplin et al. [35]	2000	Prospective	1	46.0		1/0	NS	Inoperable	1 (100%)	111In	2.8 (mean)	4	11.4	NS
Valkema et al. [36]	2002	Prospective	6	55.0 (57.5) (28–77)		NS	NS	Progressive	6 (100%)	111In	NS	NS	42.5 (28.7) (5.8–87.3)	9.5 (5.6) (1–27)
Buscombe et al. [37]	2003	Retrospective	2	52.0 (52.0) (46–58)		1/1	NS	Progressive	2 (100%)	111In	NS	3–4	10.9 (10.9) (10.5–11.4)	18 (18) (18–18)
Pasieka et al. [38]	2004	NS	1	46.0		0/1	NS	Progressive	1 (100%)	111In	5.9 (mean)	2	11.8	9
-	2000–2004	-	10	-		2/2	-	Progressive or advanced or inoperable 10/10 (100.0%)	10/10 (100.0%)	-	-	-	-	-
**References**	**Radiographic Response**						**Biochemical Response**			**Treatment Outcome**		**Treatment Toxicity**		
**Authors, Year**	**Subjects** **Suitable for Evaluation *n***	**Response Criteria**	**PD *n* (%)**	**SD *n* (%)**	**PR *n* (%)**	**CR *n* (%)**	**Subjects** **Suitable for Evaluation *n***	**Response Criteria (Calcitonin, CT)**	**Response**	**PFS Months**	**OS Months**	**Discontinuation *n* (%)**	**Grade III/IV Nephrotoxicity *n* (%)**	**Grade III/IV Haemotoxicity *n* (%)**
Caplin et al., 2000 [35]	0	-	-	-	-	-	1	Normalisation	1 (100%)	NS	NS	0 (0%)	0 (0%)	0 (0%)
Valkema et al., 2002 [36]	5	SWOG	3 (60%)	2 (40%)	0 (0%)	0 (0%)	5	Markers (NS criteria) PD SD	2 (40%) 3 (60%)	NS	NS	NS	0 (0%)	NS
Buscombe et al., 2003 [37]	2	RECIST	0 (0%)	0 (0%)	0 (0%)	2 (100%)	0	-	-	NS	NS	0 (0%)	0 (0%)	0 (0%)
Pasieka et al., 2004 [38]	1	“WHO modified” **	1 (100%)	0 (0%)	0 (0%)	0 (0%)	1	PD (increase ≥ 25% in basal value) NO RESPONSE (none of the others) PR (decrease ≥ 50% in basal value) CR (normalisation)	1 (100%) 0 (0%) 0 (0%) 0 (0%)	NS	NS	0 (0%)	0 (0%)	0 (0%)
-	8	-	4 (50.0%)	2 (25.0%)	0 (0%)	2 (25.0%)	7	-	-	-	-	0/4 (0.0%)	0/10 (0.0%)	0/4 (0.0%)

PRRT: peptide receptor radionuclide therapy, SSR: somatostatin receptor, 111In: 111 indium; SSA: somatostatin analogue, MTC: medullary thyroid carcinoma, F: female, M: male, GBq: GigaBecquerel, NS: not specified, PD: progressive disease, SD: stable disease, PR: partial response, CR: complete response, PFS: progression-free survival, OS: overall survival, SWOG: Southwest Oncology Group, RECIST: Response Evaluation Criteria In Solid Tumours, and WHO: World Health Organization. ** “WHO modified”, PD described as the “appearance of new lesions or increase of 25% or more in size of existing lesions”.

**Table 3 jcm-09-03507-t003:** PRRT targeting CCK2Rs in MTC.

References				Demographics			Disease Data				Treatment Schedule and Follow-Up					
Authors	Year	Type of Study	Subjects *n*	Age-Years Mean (Median) (Range)		F/M *n*	Mutation Status *n*	Disease Status at Baseline		Subjects with Metastases %	Radioisotope	Dose/Cycle mCi/m^2^	Cycles *n*	Cumulative Dose-GBq Mean (Median) (Range)		Follow-Up Months Mean (Median) (Range)
Behr and Behe [17]	2002	NS	8	NS		NS	NS	Progressive and advanced		8 (100%)	90Y	30–50	≤ 4	NS		NS
**References**	**Radiographic Response**							**Biochemical Response**			**Treatment Outcome**			**Treatment Toxicity**		
**Authors, Year**	**Subjects** **Suitable for Evaluation *n***		**Response Criteria**	**PD *n* (%)**	**SD *n* (%)**	**PR *n* (%)**	**CR *n* (%)**	**Subjects** **Suitable for Evaluation *n***	**Response Criteria (Calcitonin, CT)**	**Response**	**PFS Months**		**OS Months**	**Discontinuation *n* (%)**	**Grade III/IV Nephrotoxicity *n* (%)**	**Grade III/IV Haemotoxicity *n* (%)**
Behr and Behe, 2002 [17]	NS		NS	-	-	-	-	0	-	-	NS		NS	0 (0%)	0 (0%)	3 (38%)

PRRT: peptide receptor radionuclide therapy, CCK2R: cholecystokinin 2 receptor, MTC: medullary thyroid carcinoma, F: female, M: male, GBq: GigaBecquerel; NS: not specified, 90Y: 90 yttrium; PD: progressive disease, SD: stable disease, PR: partial response, CR: complete response, PFS: progression-free survival, and OS: overall survival.

**Table 4 jcm-09-03507-t004:** PRRT targeting SSRs and CCK2Rs in MTC: registered clinical trials.

ClinicalTrials.gov Identifier	Radiopharmaceutical under Study	Trial Name	Study Phase	Medical Condition under Investigation	Assigned Intervention	Primary Outcome	Estimated Enrolment, *n*	Estimated Study Completion Date	Trial Status
NCT00002947	111In-DTPA-D-Phe-Octreotide	A Phase I Study of (111In-DTPA-D-Phe)-Octreotide in Patients with Refractory Malignancies Expressing Somatostatin Receptors	Phase 1	Refractory Malignancies Expressing Somatostatin Receptors *	4 cycled of 111In-DTPA-D-Phe-Octreotide	Determine the MTD, toxic effects, and the preliminary antitumor activity of indium in 111 pentetreotide.	35	NA	Terminated (reason: NS)
NCT03647657	177Lu-PP-F11N	177Lu-PP-F11N in Combination With Sacubitril for Receptor Targeted Therapy and Imaging of Metastatic Thyroid Cancer (Lumed Phase 0/B)	Early Phase 1	Medullary Thyroid Cancer	Intravenous application of 2 × 1 GBq 177Lu-PP-F11N with and without co-medication with Sacubitril	Tumour radiation doses [time frame: measurement up to 72 h after each injection of 177Lu-PP-F11N]; evaluation of the radiation doses in tumour tissue from MTC after injection of 177Lu-PP-F11N alone and in combination with Sacubitril	8	October 2021	Recruiting
NCT02088645	177Lu-PP-F11N	177Lu-PP-F11N for Receptor Targeted Therapy and Imaging (Theranostics) of Metastatic Medullary Thyroid Cancer—a Pilot and a Phase I Study	Phase 1 **	Medullary Thyroid Cancer	Phase 1: intravenous application of 3 x max. 15 GBq 177Lu-PP-F11N. All patients with or without Physiogel, depending on the results of the phase 0 study **	Phase 1: MTD (time frame: up to 9 months) **	Phase 1: 12–18 **	March 2022	Recruiting
NCT04106843	177-Lu-DOTA-Tyr3-Octreotate	A Phase II Study to Evaluate the Effects of 177Lu-DOTATATE in Patients with Unresectable and Progressive Rare Metastatic Endocrine Carcinomas: Medullary Thyroid Cancer, Parathyroid Carcinoma, Pituitary Carcinoma, and Malignant Pheochromocytoma/Paraganglioma	Phase 2	Medullary Thyroid Cancer, Parathyroid Carcinoma, Pituitary Carcinoma, and Malignant Pheochromocytoma/Paraganglioma	177Lu-DOTATATE intravenously (IV) over 30 min every 8–16 weeks. Treatment continues for up to 52 weeks in the absence of disease progression or unacceptable toxicity	Radiographic Response (time frame: up to 52 weeks)	50	January 2022	Not yet recruiting

PRRT: peptide receptor radionuclide therapy, SSR: somatostatin receptor, CCK2R: cholecystokinin 2 receptor, MTC: medullary thyroid carcinoma, MTD: maximum tolerated dose, DTPA: diethylenetriaminepentacetic acid, and NA: not applicable. * Including head and neck cancers. ** The study includes a phase 0 whose results have already been published [20].

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
