# Peer review of "Advances in the Management of Medullary Thyroid Carcinoma: Focus on Peptide Receptor Radionuclide Therapy"

_jcm, 2020, doi:10.3390/jcm9113507_

Round 1

Reviewer 1 Report

In the review article

Advances in the management of medullary thyroid carcinoma: focus on peptide receptor radionuclide therapy”,

the authors describe very well the recent state of literature for the treatment of medullary thyroid carcinoma with special focus on PRRT. The manuscript is very well written and good to read.

The recent literature is included; however, I am missing comments and references to national and international guidelines. Can you please extent the manuscript according to this and discuss the part of PRRT in guidelines for the treatment of medullary thyroid carcinoma.

Reviewer 2 Report

This review contains valuable information about new therapeutic options for relatively rare diseases, including MTC and NETs.

 I would suggest the following edits:

-The overall format or structure of the paper is difficult to discern, particularly the Results section. Suggest explaining this in the opening paragraph of Results, i.e. what do Sections 1,2,3 represent/why is it organized in this fashion?

-The information detailed in the text of the Results is very nicely represented in the Tables. I would recommend eliminating most of the text from the Results section (details about Demographics, Disease data, Treatment, Response, etc) and simply referencing the Tables. This would make the text much more readable 

-Suggest also shortening the Discussion section. Would highlight the key points of the more important studies, rather than review the data again, in order to emphasize the information you want the reader to take away. For example, the final paragraph about PRRT targeting CCK2Rs does this and presents the idea of a promising future treatment for advanced MTC. However, in the earlier sections of the Discussion, the key points about PRRT targeting SSRs gets lost in the recap of the Results, which is not needed.

Specific suggestions:

p. 1, line 42-44 -this sentence (language) is unclear

p. 2, line 48-49-remove "effective in progressive disease", as this was already stated

p. 2, line 50-explain/define grade III-IV adverse events; change "further" to "other" systemic treatment options

p.2 line 51-should you say NEN or NET, since these therapies don't apply to benign NENs?

p.2, line 53-change to "use has been limited in MTC"

p.2, line 55-"SSR" not "the SSRs"

p.2 line 79-unclear what is meant by "section published papers"-do you meant the sections of your paper describing published papers?

p.2, line 83,87-change "detected" to "found"

p.3, line 95-change "only in" to "in only"

p.3, line 109-112-would highlight the Response data

p.3, line 113-"calcitonin measurement" not "evaluation of calcitonin"

p.3, line 117 -Why group <50% reduction and increase in calcitonin together? suggest separating out the data for these two groups if possible

p.10, line 196-consider adding to RCTs "and Ongoing clinical trials"

p.10, line 199-204-suggest omitting the study that was terminated as I don't think it adds anything (you also reference it in the Discussion). You could put the information in an Appendix 

p.10, line 218-define the abbreviation MTD here

p.12, line 245 -"has demonstrated" not "had demonstrated"

p.12, line 273-change "had an" to "was of" older age

p.13, line 281,282-change "a longer survival" to "longer survival"

p.13, line 284-not just tumor recurrence but progression?

p.13, line 287-change "hinder" to "have hindered"

p.13, line 292-293 - this is an Important point

p.13, line 299-change "in none of cases" to "in no cases"

p.13, line 303-change "with as consequence" to "with the consequence of"

p. 13, line 304-306-change "could" to "may" and suggest changing "it is plausible" to "It is unknown whether late renal damage..." and omit the word "later"

p.13, line 307-reword the sentence, suggest "The limitations of this Review study about data on PRRT targeting SSRs in MTC are: "

p.13, line 311-change "size sample" to "sample size" and after "length" add "of follow up"

p.13, line 313-change "having as" to "with the" primary outcome

p.13, line 317- 319-change "has" to "have" and "regarded the preclinical setting" to "were preclinical trials"

p.13, line 322-change "courts" to "cohorts" and "about this topic is the study" to landmark "study in this area was"

p. 14, line 333-change "concerns" to "problems"

p.14, line 339-change "radiopeptides" to "radiopeptide"

p.14, line 342-change "shown" to "evidenced" and add "to" after the word "pertaining"

p.14, line 342-change "less" to "lesser" and add the word "and" after "new" and before "efficient"

-Tables 1, 2, 3- change the spelling of "Cicles" to "Cycles"

-some of the tables are busy and are missing a lot of data; can you eliminate the columns where the data is not available? Can you change "NS" to "-" which makes the table a little less busy? Can you eliminate the less important columns and put those in an Appendix that you reference in the text, e.g. "Suitable Subjects", "Treatment Outcome" or "Treatment Toxicity"? Smaller tables may be a better way to present the data more concisely

-My MAJOR suggestion would be to eliminate most of the test from Results and briefly refer to the Tables, ie pages 2,3,6,8,10

Reviewer 3 Report

The authors present a review on the use of PRRT in MTC. In my view, the manuscript is of interest to the clinical community and summarizes well valuable information; therefore I believe it would be of interest to the readership of JCM. However, there are some flaws that need to be addressed. The manuscript would benefit from further revision following the methodology of a systematic review and also clarifications with respect to its limitations. 

  1. Please provide a PRISMA flow chart of the selected studies and follow the the PRISMA guidelines in the results section. Please consider searching through EMBASE and Cochrane Central also.
  2. Please use standard nomenclature to  provide a quality/risk of bias assessment of the included studies, i.e. Newcastle-Ottawa Scale for cohort studies or other system depending on the type of the included studies.
  3. Page 3, line 95-97: please clarify if the study cited here also includes patients without MTC, i.e. other NENs as in MEN1.
  4. Page 10, line 197-220. This part should be omitted from the results section, as there are no data on PRRT eficacy, toxicity etc. Please consider a shorter version of this that could be added in the discussion section.
  5. Page 12, line 226-258. This part is too general and should be omitted as it does not add anything to the discussion of the paper. Please consider a brief summary of the findings of the Review as the first paragraph of the discussion section.
  6. Page 13, line 275-278. Please expand on combining chemotherapy with PRRT as this is currently a novel strategy in high grade mestastic NEN.
  7. Please elaborate briefly in the discussion section on the PRRT efficacy in MTC in relation to that in other NENs.
  8. Please elaborate briefly in the discussion section on the PRRT efficacy and toxicity in relation to TKIs. What would be the appropriate sequencing when considering TKIs and PRRT in MTC patients?
  9. Please expand the paragraph with the conclusions of the Review, including safety aspects.

Round 2

Reviewer 2 Report

I think you did a great job with reorganizing the paper and highlighting the important points. It reads well and is well presented.

The following are some additional editing suggestions, most of which are related to language or clarity:

p.2, line 25 --insert "a" after "used as" and before "therapeutic option"

p.2, line 33--would change "expresses also" to "also expresses"

p.2, line 41--would change "a subgroup of patients shows" to "in a subgroup of patients, the tumor shows"

p.3, line 52--change "has been" to "was"

p.3 line 55, 59--change "Somatostatin receptors (SSR)s" to "Somatostatin receptor (SSR)" expression and "cholecystokinin receptors (CCK2R)s" to "cholecystokinin receptor (CCK2R)" I think in most cases throughout the paper, the receptor expression should be referred to as singular, not plural

p.3, line 68--eliminate "to" and change to "give the scientific community"

p.4, line 96--eliminate "of" and change to "low dose oral capecitabine"

p.4, lines 99-100, 116,119--define the abbreviations PD, PR,PFS and OS if this is the first time you are using them in the paper

p.4, line 120--consider clarifying this statement about calcitonin doubling time, as it is not clear whether you are referring to an effect of the PRRT (the statement may also apply to MTC patients whose tumors have longer doubling time).

p.9, line 186--add "were" after "most of which"

p.9, line 199, 200--omit "a" before "calcitonin assessment" and change "assess" to "indicate" treatment response, omit "a" before "good sensitivity"

p.9, line 217--change "SSRs-positive well-differentiated" to "SSR-positive, well-differentiated"

p.9, lines 214-223--this is a good paragraph

p.9, lines 228,229--sentence is unclear; suggest rewording to "As with other NENs, 111In-DTPA-Octreotide was the first therapeutic radiopharmaceutical used as PRRT in MTC. "

p.9, line 233--change "early terminated" to "terminated early"

p.10, line 234--change "various" to "as follows"

p.10, line 237--add "a" after "in" and before "few"

p.10, line 240--change "having as" to "with the" and change "the" to "of" before tumor

p.10,line 248--add "a" after "Only"

p.10, line 250--change "who" to "that"

p.10,line 251-253--sentence beginning "Normal organ uptake..." is unclear. Would reword it and emphasize the issue of kidney uptake

p.10,line 263--change "to reduce" to "of reducing" and "improve" to "improving"

p.10,line 267--add "the" before "major enzymes"

Reviewer 3 Report

The manus has been significantly improved following the Reviewer's suggestions. Good luck.
